# Challenges and Open Problems of Legal Document Anonymization

Gergely Márk Csányi [1,2], Dániel Nagy [1], Renátó Vági [1,3], János Pál Vadász [1,4] and Tamás Orosz [1,*]

1   MONTANA Knowledge Management Ltd., H-1097 Budapest, Hungary; csanyi.gergely@montana.hu (G.M.C.); nagy.daniel@montana.hu (D.N.); vagi.renato@montana.hu (R.V.); vadasz.pal@montana.hu (J.P.V.)
2   Department of Electric Power Engineering, Budapest University of Technology and Economics, H-1111 Budapest, Hungary
3   Doctoral School of Law, Eötvös Loránd University, H-1053 Budapest, Hungary
4   Institute of the Information Society, National University of Public Service, H-1083 Budapest, Hungary
*   Correspondence: orosz.tamas@montana.hu

**Abstract:** Data sharing is a central aspect of judicial systems. The openly accessible documents can make the judiciary system more transparent. On the other hand, the published legal documents can contain much sensitive information about the involved persons or companies. For this reason, the anonymization of these documents is obligatory to prevent privacy breaches. General Data Protection Regulation (GDPR) and other modern privacy-protecting regulations have strict definitions of private data containing direct and indirect identifiers. In legal documents, there is a wide range of attributes regarding the involved parties. Moreover, legal documents can contain additional information about the relations between the involved parties and rare events. Hence, the personal data can be represented by a sparse matrix of these attributes. The application of Named Entity Recognition methods is essential for a fair anonymization process but is not enough. Machine learning-based methods should be used together with anonymization models, such as differential privacy, to reduce re-identification risk. On the other hand, the information content (utility) of the text should be preserved. This paper aims to summarize and highlight the open and symmetrical problems from the fields of structured and unstructured text anonymization. The possible methods for anonymizing legal documents discussed and illustrated by case studies from the Hungarian legal practice.

**Keywords:** data mining; text mining; text recognition; machine learning; knowledge engineering

## 1. Introduction

Digitalization of judicial systems is an important goal of the European Union [1]. Sharing and making court decisions and different legal documents accessible online is a crucial part of this intention. These public databases can make the administration of justice more transparent. These openly accessible documents can also help decision-making processes and research for the legal practice [2–6]. Many novel databases provide easy access to court decisions and legal documents. These databases do not only share the digitized version of the documents, but also use different machine learning methodologies to find the connecting documents and the most relevant keywords of the documents. Some novel databases go further and connect these legal documents with other databases and provide them as a linked data service [4]. The databases built on published court decisions are very attractive for a wide range of professionals in the legal information market in order to reduce the research time.

However, this openly published data can contain much sensitive personal information which should not be published [7]. The current anonymization practice of these legal documents means the masking or removing the names or other direct identifiers from these documents [8]. Many current research projects have been published in this topic,

which shows how state-of-the-art Named Entity Recognition (NER) methods can be used to achieve an automatized process, which can reduce the necessary human work, which can take up to 40 min for a single document [2,9,10].

However, these solutions result the simple pseudonymization of these documents only. General Data Protection Regulation (GDPR) makes a difference between anonymization and pseudonymization. The main difference between these methods is that during the anonymization process the anonymized data is modified irreversibly (see Figure 1) [11–13], while pseudonymization processes cannot prevent privacy breaches, as shown in the literature of medical data anonymization [14–16].

The most sophisticated tools in the legal document anonymization domain (e.g., ANOPPI and Finlex [2,5,10,17]) use state-of-the-art machine learning-based Named Entity Recognition (NER) technology to find and replace direct identifiers in judicial texts. However, without the deeper statistical analysis of the possible pseudo-identifiers, these applications cannot make an irreversible anonymization on the documents [18].

In 2019, a group of researchers carried out a linking attack against anonymized legal cases in Switzerland. They published a study where they presented that using artificial intelligence methods with big data collected from other publicly available databases, they could re-identify 84% of the people, being anonymized in this database, in less than an hour [19]. This can happen because a legal document contains a great deal of microdata from the involved participants, which can be used as quasi or pseudo identifiers to re-identify the participants by using third party databases (Figure 2) [20–22].

Furthermore, there are available methods for anonymizing medical data for research purposes [16,23] but, these anonymization methodologies have been developed for structured data, where every record contains the same kind of attributes and removing the items does not have an effect of the text understandability. For this reason, this topic is asymmetric regarding the significant difference in the behaviour of structured and unstructured data.

In case of a legal document, the processed data is unstructured text; therefore, the extracted data can be represented in a sparse matrix of different attributes [24]. It is also important that the anonymized text should remain easy to read after the pseudonymization process [25]. Let us examine the case of the following example sentence: "John and Julie went on holiday to Barbados in August 2016". Simple masking replaces every name, date and place in the text with "XXX". This results in losing essential text properties which are needed for the utilization of the text [26]. Furthermore, if a person is referred as "Iron Lady", the NER-based anonymization tools will not recognize it as a direct identifier; however, the reader will probably know from the context that Margaret Thatcher is the referred person.

Mozes and Kleinberg proposed a TILD methodology (TILD is an acronym from Technical tool evaluation, Information loss and De-anonymization) [27] for anonymizing texts. They proposed to analyze three objectives during the anonymization process together. These three critical aspects are how well the system detects the pseudo and direct identifiers, how much information is lost by removing the different parts of the sentence and to assess the possible data breaches.

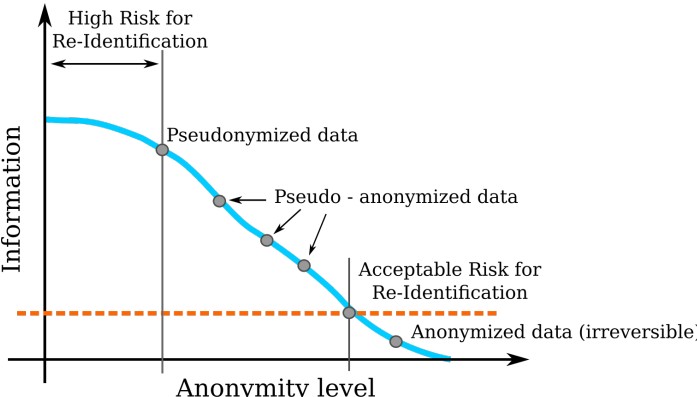

**Figure 1.** The difference between the anonymous and the pseudo-anonymized data. The red line represents the threshold, where the re-identification of the anonymized or anonymous data is impossible.

**Theorem 1** (Pseudonymization, GDPR [28]). *"The processing of personal data in such a manner that the personal data can no longer be attributed to a specific data subject without the use of additional information, provided that such additional information is kept separately and is subject to technical and organisational measures to ensure that the personal data are not attributed to an identified or identifiable natural person".*

**Theorem 2** (Personal Data, GDPR [28]). *"Personal data is any data that could possibly related to a person directly or indirectly (Opinion 4/2007 on the concept of personal data)".*

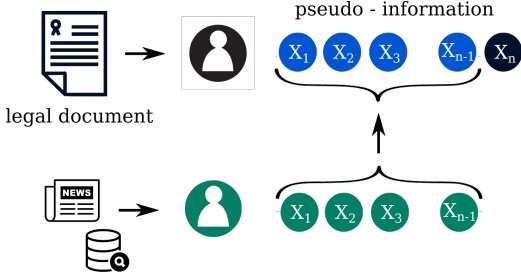

**Figure 2.** The image shows the found pseudo or quasi-identifiers in the text $(x_1...x_n)$ the role of anonymization and privacy-preserving data publishing, and a possible linking attack, which uses publicly available data from newspapers and other public databases.

The main goal of the paper is to provide a thorough insight on the flaws of the current practice of judicial text anonymization processes. The current anonymization practice in many European Union countries means the masking of the names and other direct identifiers of the involved persons. This process does not fulfill the requirements of the General Data Protection Regulation. The first part of the paper summarizes the existing data anonymization techniques, which mostly support anonymizing structured data. The goal of the second part of the paper is to illustrate and highlight the importance of this topic. These examples come from the database of the decisions of the Hungarian Court [29]. These examples show that mathematical statistical analysis is important in filtering those unique events, that may serve as a primary identifier (e.g., the surgeon amputates the wrong leg). Moreover, those applications and services, which link the legal documents together with other databases, need a special care to consider the GDPR recommendations.

## 2. Privacy and Anonymization

The apparent goal of the data anonymization or de-identification process is to remove all of the involved individuals' direct identifiers, such as names, addresses, phone

numbers, etc. [16]. This step might seem obvious; however, it is easy to defeat this kind of anonymization by linking attacks or more sophisticated attacking models [30]. These algorithms collect the text's auxiliary information and link it with public, not anonymous records [31–33], as shown in Figure 2.

**Theorem 3** (Pseudo or quasi-identifiers [34,35])**.** *Attributes that do not identify a person directly but by linking to other information sources (e.g., publicly available databases, news sources) that could be used to re-identify people are called pseudo or quasi-identifiers. e.g., date of specific events (death, birth, discharge from a hospital etc.), postal codes, sex, ethnicity etc. [36–39].*

Sweeney made a famous linking attack on the set of public health records collected by The National Association of Health Data Organizations (NAHDO) in many states where they had legislative mandates to collect hospital-level data (Figure 3). These data collections were anonymized but contained several pseudo-identifiers (ZIP code, gender, ethnicity, etc.). She bought two voter registration lists from Cambridge, Massachusetts. These voter registration lists contained enough pseudo-identifiers to be linked with medical data records [31,40]. Three pieces of pseudo-information (ZIP code, gender, and date of birth) were enough to link the two databases, and Sweeney was able to re-identify about 88% of the people. She also found that about 87% of the population in the United States could be identified by using simple demographic attributes (5-digit ZIP, gender, date of birth) and about 53% when only place, gender, and date of birth is known [22,40].

Another good example would be the case study about the Netflix Prize dataset. Netflix, at that time renting DVDs online, published a dataset and offered USD 1 million as a prize to improve their movie recommendation system [20,21,41]. The dataset contained ratings for different movies and the exact dates when these ratings were made. The studies of Narayanan and Shmatikov have shown that users can be identified with about 95% probability only by knowing one's ratings for eight movies (out of which six is known correctly) and knowing the date of publishing within a 2-week-long interval [20,21]. Although, at first sight, knowing someone's taste might not seem to be dangerous at all, and in the vast majority of the cases this would be true, political orientation or religious views might be just two examples for possibly sensitive information that could be learned from one's rating history and surely not everyone would agree to share these pieces of information with the whole world [21].

A similar privacy breach on unstructured data, namely on query log data, has been reported in [42].

Figure 2 shows a basic attack model against an anonymized legal case. Suppose the aim of the curator is to preserve privacy for n pseudo-information of the legal document. However, an attacker has information about $n - 1$ pseudo information of an individual except the $n$th record, denoted by $X_n$. This information of the attacker (the $n - 1$ records) is also known as the background information. By making a query on the dataset of legal documents, the attacker can learn the aggregate information about n records. Hence, by comparing the query result and the background information, the attacker can easily identify the additional information of the record $n$. Having spotted the additional information, the attacker could move on to other datasets, performing the same operation until the victim can be re-identified. The aim of differential privacy is to protect against these kind of attacks [30,43].

**Theorem 4** (Equivalence class [44])**.** *"All data records that share the same quasi-identifiers form an equivalence class". For example, all companies founded on 2nd July 2018 form an equivalence class.*

**Theorem 5** (Differential privacy [30])**.** *"A mechanism M satisfies ε-differential privacy if, for any datasets x and y differing only on the data of a single individual and any potential outcome $\hat{q}$:*

$$\mathbb{P}[\mathcal{M}(x) = \hat{q}] \leq e^{\epsilon} \cdot \mathbb{P}[\mathcal{M}(y) = \hat{q}], \tag{1}$$

*where the $\epsilon$ parameter represents the the bound of the privacy loss and its value can be selected between 0 and 1. Where 0 represents that there is no privacy loss, the result of the two examined methodology is the same, no extra information is revealed from the selected individual".*

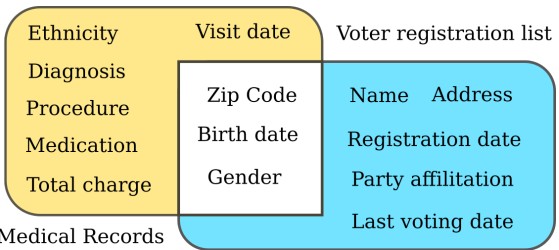

**Figure 3.** The image shows Sweeney's linking attack. Three pseudo identifiers were enough to re-identify the private data of many individuals.

Differential privacy is a strict, robust, and formal definition of data protection, which ensures the following three criteria: post-processing, composition, and group-privacy of the data. Post-processing means that the analysis of the published information or linking this microdata with other databases results in a set of data that satisfies the criteria of differential privacy. Therefore, it protects from the successful reconstruction or tracing from this data. Composition and group-privacy also protects privacy, i.e., when the information is shared in multiple places or with multiple individuals [30,45].

*2.1. Privacy Models*

Ensuring privacy while maintaining the utility of data are two directions contradicting each other. To satisfy both needs, several privacy models have been introduced. These models use generalization or suppression (i.e., deleting) of attributes in records to reach the defined privacy [16,46].

These models distinguish three types of data: direct identifiers (e.g., full name, social security number, etc.), quasi-identifiers (e.g., age, job, date of birth, etc.), and confidential attributes (e.g., religion, specific disease, salary, etc.) [47]. Privacy models assume that the directly identifying attributes have been removed and focus on quasi-identifiers and confidential attributes. There are many privacy methods represented in the literature [16,48–50]. In this section, we present some widely used methods: k-anonymity [51], l-diversity [52], and t-closeness [53].

2.1.1. k-Anonymity

A widely known approach is k-anonymity, which has been introduced by Samarati and Sweeney [51,54–56]. The formal definition can be seen in Definition 6.

**Theorem 6** (k-anonymity [51,54–56])**.** *"A dataset satisfies k-anonymity for k > 1 when at least k records exist in the dataset for each combination of quasi-identifiers".*

This privacy model does not ensure privacy in itself. K-anonymity is able to prevent identity disclosure, which makes impossible the exact mapping of the k-anonymized record to the original dataset. However, it is prone against attribute disclosure, e.g., if k pieces of k-anonymized records share the same confidential attributes [47]. As an example, if there are k number of records identified by the following attributes: Age = 55, Height = 185 cm, Sex = Male, but all records share the same sensitive information, e.g., all of them have AIDS.

2.1.2. l-Diversity

An approach to solve attribute disclosure is l-diversity [52]. The formal definition can be seen in Definition 7.

**Theorem 7** (l-diversity [47,53]). *"An equivalence class has l-diversity if there are at least l "well-represented" values for the sensitive attribute. A table has l-diversity if every equivalence class of the table has l-diversity".*

The term "well-represented" is not an exact definition. Machanavajjhala et al. have given the following interpretations: distinct l-diversity, entropy l-diversity, recursive (c,l)-diversity [52].

However, Li et al. [53] pointed out that l-diversity cannot prevent disclosure in case of skewness and similarity attacks, and that achieving l-diversity may be difficult and unnecessary [47].

*Similarity attack* [47] can be performed when the group of sensitive attributes fulfill the criterion of l-diversity but are semantically similar, e.g., in a 3-diverse medical dataset where disease is a sensitive attribute, the values are (lung cancer, stomach cancer, skin cancer). Despite meeting the requirement of 3-diversity, it is possible to learn that someone has cancer.

*Skewness attack* [47] can happen when the overall distribution is skewed, in which case l-diversity cannot prevent attribute disclosure. Consider a database of 10,000 test results for a disease where being positive is sensitive information, and there are 1% positive tests. An equivalence class having 24 positive records and only one negative record would meet the criteria of distinct 2-diversity and would have higher Entropy l-diversity than the whole dataset, although anyone in the equivalence class is 96% positive rather than 1%. Oddly, the same l-diversity could be calculated when the equivalence class contains only one positive and 24 negative members, although the risks are highly different.

### 2.1.3. t-Closeness

Li et al. in [53] defined t-closeness (Definition 8) that provides a solution for the two vulnerabilities of l-diversity mentioned above, namely the skewness and similarity attacks.

**Theorem 8** (t-closeness [53]). *"An equivalence class is said to have t-closeness if the distance between the distribution of a sensitive attribute in this class and the distribution of the attribute in the whole table is no more than a threshold t. A table is said to have t-closeness if all equivalence classes have t-closeness".*

Li et al., in their work, used the Earth Mover's Distance [57] as distance metric to calculate t-closeness [53]. However, the authors claim that using other distance metrics (e.g., cosine-distance, Euclidean-distance etc.) is also possible.

Domingo et al., in [47], criticizes that although in [53] several ways have been represented to check t-closeness, no computational procedure has been given to enforce this property. Moreover, if such a procedure was available, it would cause huge harm to data utility since t-closeness destroys the correlations between quasi-identifiers and confidential attributes. The only way to prevent damage to data utility is to increase threshold t, hence relaxing t-closeness.

### 2.2. Available Tools, Solutions

The introduction of GDPR has increased the number of pseudonymization software or solution available. In this section, we provide a short overview regarding solutions that are connected or could be connected somehow to the anonymization in legal domain.

Vico and Calegari [58] presented a general solution for anonymizing a document in any domain and tested its functionality on legal documents, although no quantitative validation was presented in their paper. The whole solution is more akin to a generic flowchart that could be applied to any type of document. The backbone of the method is a Named Entity Recognition model. The new idea that the paper brought in is that from the results of the extraction, the entities belonging to the same person or location were assigned to each other by means of clustering, e.g., if the full name John Doe was mentioned once in the text, and afterward it is also referred to somewhere as J. Doe or John D., the entities

were considered to be the same. The found entities have been then modified to generic terms. The drawback of this solution is that it does not differentiate between direct and quasi-identifiers, it only focuses on extracting direct identifiers, thus risk analysis of the remaining quasi-identifiers is also missing.

Povlsen et al. adopted a Danish NER solution to the legal domain, based on hand-crafted grammar rules, gazetteers, and lists of domain specific named entities [9]. The solution was tested on 16 pages of legal content containing 30 entities, that is a small dataset for testing. Moreover, the authors focused on identifying direct identifiers not taking quasi-identifiers into consideration during their anonymization process.

NETANOS, an open-source anonymization tool, focuses on context-preserving anonymization to maintain readability of texts. This is practically carried out by replacing the entities by their types, e.g., "John went to London with Mary". is replaced by "[PER_1] went to [LOC_1] with [PER_2]". The study offers a comparison between manual anonymization, NETANOS anonymization and UK Data Service (https://bitbucket.org/ukda/ukds.tools.textanonhelper/wiki/Home) (accessed on 1 May 2021) anonymization techniques involving hundreds of people. Authors claim that their software achieves almost the same level in possibility of re-identification as manual anonymization [25]. However, this solution also takes direct identifiers into consideration, without paying attention for quasi-identifiers.

There are many tools available for anonymization of medical records and health data, such as the UTD Anonymization Toolbox [59], $\mu$-AND [60], the Cornell Anonymization Toolkit [61], TIAMAT [62], Anamnesia [23] or SECRETA [63]. These solutions are able to fulfill privacy criteria defined by the user automatically [16]. However, the shortcoming of these solutions is that they often support only a limited number of privacy and data transformation models [16]. ARX [64] is an open-source anonymization tool, which supports a wide range of anonymization techniques, such as k-anonymity, l-diversity, etc. These techniques were developed to provide a flexible and semi-automatic solution for anonymization of data tables [16,46,64,65]. The tool was developed for medical data that is in a database format; therefore, these software solutions cannot contain methodologies for natural language processing of unstructured texts.

HIDE is a tool developed to anonymize health data [66,67]. The tool takes into account that a significant part of health data exists in unstructured text form, e.g., clinical notes, radiology or pathology reports, and discharge summaries and extracts direct identifiers (e.g., patient name, address, etc.), and quasi-identifiers (e.g., age, zip code, etc.) and sensitive information (e.g., disease type) from the unstructured text. Since a person can have multiple health scans, the device tries to attach the information extracted from the scanned document to people existing in the database. The database thus expanded allows HIDE to perform anonymization procedures such as k-anonymity, t-closeness, l-diversity on the whole database that is a traditional relational database. Extraction is based on a Conditional random Field (CRF) NER model. The wide range of extracted data and organizing the documents to a database makes HIDE an outstanding anonymization tool. However, the solution was implemented to tackle medical documents, not legal ones, so inherently misses entities characteristic for the legal domain (e.g., events, multiple sides etc.).

ANOPPI [2], is an automatic or semi-automatic pseudonymization service for Finnish court judgments. It uses state-of-the-art, BERT-based [18] NER models alongside rule-based solutions to retrieve as much direct identifiers as possible from legal texts. However, the solution does not take into consideration the fact that other quasi-identifiers, e.g., events in themselves can be direct identifiers, or a small set of quasi-identifiers can lead to a privacy breach as we show in Section 5.4. The tool emphasizes the importance of utilizing the legal text after pseudonymization has been carried out, since Finnish is a highly inflected language (similar to Hungarian). The tool performs morphological analysis in order to apply the correct inflected form for the pseudonym, hence improving readability of pseudonymized text. The aim of the ANOPPI project is to create a general purpose anonymization tool, by removing direct identifiers. It has been shown that methods, which remove the directly identifying attributes only (i.e., names, email addresses or personal

identification numbers) can not prevent privacy breaches [14–16] and would contradict the No Free Lunch Theorem in related fields [68–70].

## 3. Types of Privacy Attacks

Knowing the different types of privacy attacks is essential to considering and quantifying the different privacy risks. There can be different goals of the adversaries: re-identification, reconstruction or tracing of different persons, where the attacker only wants to know that the given person is connected to the given dataset or not [30]. The authors of [49,71] published three different kinds of attacker models (prosecutor, journalist and marketer), which typically considered in medical health data anonymization software solutions [16,46]. These attacking strategies assume that the adversary has different a priori information about the database or the subject of the attack. For instance, the prosecutor knows that the data about the searched person is involved in the given or the connecting cases. The journalist has some a priori information about the searched person, maybe she knows some background data, which can be linked with the microdata of the connecting cases. The marketer model assumes that the adversary has no prior knowledge, but their goal is to re-identify a large number of individuals for marketing purposes [71,72].

The previously introduced Swiss case study about re-identifying pseudonymized legal cases used a marketer strategy with a simple linking attack. The researchers wanted to re-identify as many people as they can, surprisingly, they could re-identify 84% of them within an hour) [19]. The reason for this is the presence of a large number of quasi-identifiers in legal documents. The wide range of quasi-identifiers, both in number and type, generally provide enough information for a human to de-anonymize the documents. As Mozes and Kleinberg pointed out, even a single identifying attribute can be sufficient for re-identification [27]. Since legal cases always tell the story of the two or more sides involved, the texts contain many events, people, and institutions (two sides, judges, attorneys, witnesses, etc.). A legal document contains one case, which is interpreted in at least three different ways. It is possible that each of these single interpretations cannot contain enough microdata to re-identify the persons, but the series of these interpretations serve enough information for de-anonymization (Figure 4), as it is shown in Section 5.4.

Figure 4 shows an example of how legal documents may be connected to each other or to other databases via the wide range of these quasi-identifiers. It is important to point out that the majority of the published cases are part of a "decision-chain": decisions from the first instance up to the Supreme Court are linked together. This means that there are usually three documents being linked tightly to each other. This poses a threat of de-anonymization since it is not sufficient to have two of these documents properly anonymized if the third is not properly anonymized. Hence, the integrity of a chain must be kept. The quasi-identifiers learnt from the joined documents can be used to match these data with other publicly available databases in order to de-anonymize the parties involved.

Because many novel legal databases aim to find and publish the connecting cases, these databases can increase the risk of attacks, as mentioned above [4,29]. In the case of a prosecutor attack where the adversary has a priori information about the person, linking databases can give extra information for the adversary. The main difference between the medical and the legal databases is that the database contains not only the data fields, but the context of the text can contain many quasi-identifiers, which can help the attacker gain some new information. For example, suppose the attacker wants to know for whom the orthopedist performed the surgery on the wrong side. In that case, he would only check the local journals and the homepage of the small hospital, and he will know not only the name of the doctor but also the patient's information from the text of the legal document. The connecting cases can link other persons to this document. The previously mentioned algorithms, which were developed for medical data anonymization (k-anonymity, t-closeness, and l-diversity) can help to reduce the number of the quasi-identifiers in the text. However, the over-usage of these algorithms can significantly reduce the understandability of the text. Moreover, the quasi-identifier

structure leads to a sparse matrix, not a dense as in medical text anonymization, where the methodologies mentioned above developed. Due to the asymmetries between these databases, the mathematical model of these two fields leads to different approaches.

Aa an example of a journalist attack, we can examine the case of a small district court, where there are a small number of potential criminals. If we know some marker of a criminal or a criminal group, which committed many similar cases in the area, we can connect the involved persons via the connecting legal documents. This can easily lead to a lot more sensitive information about the involved people.

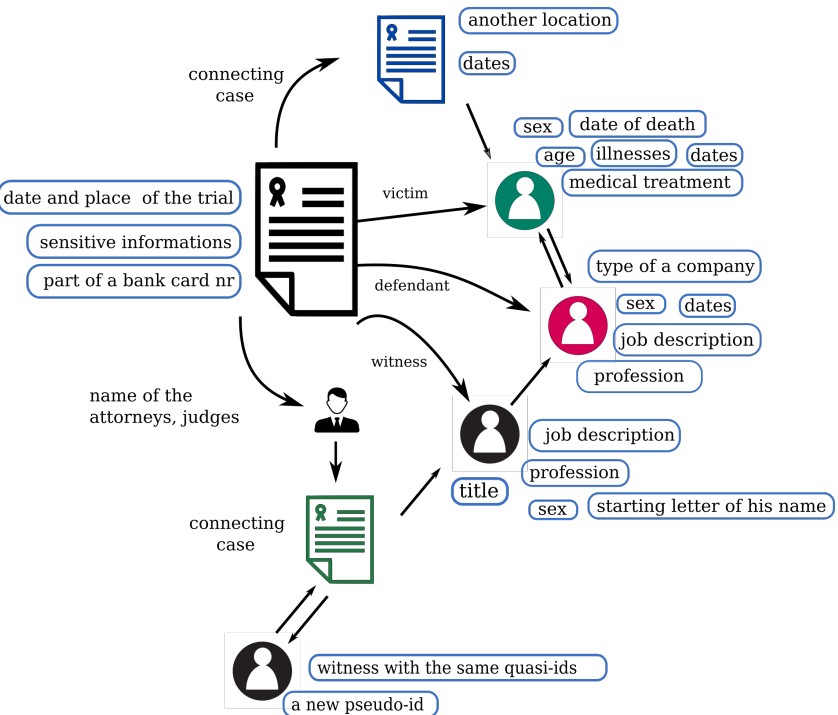

**Figure 4.** The image shows how a published legal document can connect to other legal documents directly or via the judges and attorneys, and the connections between the recognizable quasi-identifiers, which can be assigned for different people. These identifiers can be used to perform a linking attack, linking data to other databases.

## 4. Does Document Domain Matter? Differences between Medical and Legal Anonymization Tasks

At first sight, medical data and legal texts share several similarities (domain-specific language, structured content to some extent, a wide range of data types to be anonymized etc.), there are specific problems that characterize legal documents only. Medical data is often available in unstructured text format (e.g., clinical notes, radiology, and pathology reports, and discharge summaries) [66] where legal cases only exist in this form.

One main difference would be that in the majority of medical text there are at most two subjects mentioned: one is the patient, who has diseases, several IDs, job, and the other is the hospital where he or she is being treated. It is relatively rare to find data that is pointing out from this context (e.g., other people, natural formations, car plate numbers etc.). Therefore, linking the extracted entities to a specific person is obvious in medical texts [67]. Whereas in the majority of legal cases there are also two parties (plaintiff and defendant), it is not rare to have more than one person on any or even both sides. Not to mention the witnesses, companies involved, bank accounts, etc., are also mentioned in the texts.

Another significant difference is that due to the nature of legal cases, the matter of fact describes the whole story of the parties in detail and usually this part of the legal decision is full of complex quasi-identifiers. These complex quasi-identifiers mean events

or rather chains of events that could be easily used by an attacker to form further queries while attempting to link data sources to the parties of a legal decision. An example would be an extraordinary or rarely occurring event, e.g., someone was gored by a bull, died during a gland surgery or breeds limousine-type cows that could easily appear in local newspapers or publicly available databases. It is important to point out that how much a rarely occurring event makes a difference, since that type of data does not appear in tabular health data, whereas the legal documents contain many of these, especially the matter of fact part. Moreover, the definition of "personal data" in GDPR Article 4 states that these data can be considered as indirect identifiers (Regulation (EU) 2016/679 of the European Parliament and of the Council Article 4).

Emphasizing the role and importance of quasi-identifiers in anonymization is not only the result of GDPR. About 60 years ago researchers started investigating the idea whether a small number of data points about an individual can be collectively equivalent to a unique identifier even if none of these data points are unique identifiers [33,73–75].

Health data of patients are often presented in a tabular format and these data tend to share the same columns (i.e., a traditional relational database). In this dataset, the rows (records) represent data of a patient and the columns represent the attributes a patient has (e.g., sex, date of birth, profession, disease, etc.). These attributes usually do not contain many null elements, every attribute, every column are well represented. This is not the case in legal cases. If we represent the data stored in a set of cases the same way as one does with medical records, the number of attributes, i.e., columns, can be a great deal more than in a medical database, so the dimension of the records is higher. Moreover, these data may appear relatively rare in other records, hence the matrix that could be created from a set of cases is sparse (see Figure 5). The occurrence of data that fall under the regulation of GDPR is highly asymmetric in legal documents. This sparsity puts the complexity of the problem to a completely different level compared to medical records.

The right side of Figure 5 illustrates the case of a typical medical database, where every row contains the same information about an individual. This database is symmetrical because every record has the same identifiers, and we know every possible quasi-identifier in this task. The health data anonymization methodologies use this symmetry or structural regularity. In contrast, the legal documents can be very asymmetrical, hard-to-find similar structural regularities, which increases the complexity of the risk analysis of these models. The most obvious example is the role of the personal data of the dead persons in the documents. This data belongs to the GDPR regulations in some countries, but some countries do not care about the personal rights of dead persons. However, in probate proceedings, some sensitive data of the dead person can be published. If we know the link between the dead person and the plaintiff, who is, e.g., the only living relative, we can re-identify their data. A more general risk for the asymmetrical dataset is that the document contains three-three quasi-identifiers for two individuals, which is insufficient to re-identify the involved persons. However, these six identifiers can belong to six different categories (such as occupation, age, etc.), and by knowing the relation between the individuals, it can be possible to re-identify them.

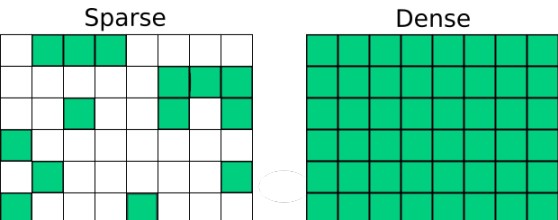

**Figure 5.** Sparse (asymmetric) and dense (symmetric) representation of data.

There is a theoretical proof for the statement that high dimensional data is vulnerable to de-anonymization [33,34,49].

Narayanan and Shmatikov claim that "Most real-world datasets containing individual transactions, preferences, and so on are sparse. Sparsity increases the probability that de-anonymization succeeds, decreases the amount of auxiliary information needed, and improves robustness to both perturbation in the data and mistakes in the auxiliary information" [21].

Thus, the task of anonymization in legal documents cannot, by its very nature, be regarded as being entirely solvable.

A good example for de-anonymizing sparse data would be the case study about the Netflix Prize dataset [20,21,41] and the America Online (AOL) search engine query log dataset [42].

Motwani and Nabar published an algorithm that achieves k-anonymity on an unstructured, non-relational dataset, namely on search engine query logs [24]. In this dataset, there was no need to distinguish between direct and quasi-identifiers. Their approach was to transform the tokens from the query logs to a relational database containing only ones and zeroes. This database had a high number of dimensions and was sparse. They then reached k-anonymity by adding and deleting as few elements as possible from each query until the k-anonymity criterion have been met. Although, in many aspects of their data, the problem itself is different from the data that could be retrieved from legal documents, the solution can be useful in reaching k-anonymity in legal documents as well.

As a consequence, performing decent anonymization in legal cases is far away from just identifying direct identifiers from the text and deleting or modifying them; in other words, to only perform Named Entity Recognition and modify the extracted entities. Nevertheless, the currently available anonymization software solutions generally represent this mindset [2,9,25].

To perform a decent anonymization on a legal document, a wide range of quasi and direct identifiers have to be recognized, in particular, the rare events mentioned in the case. After the recognition and modification of direct identifiers, the quasi-identifiers have to undergo a careful risk analysis, i.e., the risk for co-occurrence of many quasi-identifiers that can be connected to individuals has to be estimated and if needed, anonymization techniques have to be applied on them. According to our knowledge, there is currently no anonymization solution for legal texts that takes the importance of events into consideration [2,7,9,58,76,77].

Data owners have to accept that legal cases may not be de-identified [33,34,49], but not protecting these data at all is also not an acceptable option. The problem is similar to having our bicycle protected from being stolen. It is not a good tactic to park somewhere and hope for the best. Even though there is no bike lock that cannot be broken, it is still worth using at least some bike lock, because many times that is enough to deter the thief. Instead of giving up anonymization, data owners should aim to reduce the chances of an attack succeeding.

## 5. Structure and Privacy Risks in Hungarian Legal Documents

In this section, we provide an overview of the current legal system and known regulations related to anonymization in Hungary focusing on the risks that legal documents inherently contain. For other member states in the European Union, the work of van Opijnen et al., [7] provides a good general overview of data protection in the legal domain.

### 5.1. Judicial System, Regulations

Hungary has a four-tier judicial system, which consist of the district courts, the regional courts, regional courts and the Supreme Court [7].

Organisation and Administration of the Courts regulates the publication of legal documents and decisions by the Act CLXI of 2011 in Section 5.1 ("Responsibilities of Courts Relating to the Publication of Court Decisions; the Register of Court Decisions") and articles 163–165 [7].

Article 164 regulates that after the decision is rendered in writing, it shall be published by the chairman of the court in the Register of Court Decisions within thirty days (Act CLXI of 2011 on the Organisation and Administration of the Courts Article 164).

The law currently in force provides two different forms of publication. In the first case, the decisions have to be published automatically, but in the second case, publication depends on the will of the litigants. In both cases, the published documents have to be anonymized. Therefore, not all decisions are subject to the obligation to publish; decisions of lower courts are outside the scope, if the legal procedure does not reach at least the regional courts of appeals.

According to current regulations, e.g., the name of the court concerned, the name of involved judges, lawyer acting as an agent, the defense counsel or administrative organizations (e.g., National Tax and Customs Administration), or the authors of certain scientific publications are to be anonymized [7,8]. Sensitive data have to be deleted in a way that the deletion does not change the established facts of the case. Instead of the names of the persons covered by the decision, the name corresponding to their role in the procedure shall be used; instead of the names used to identify the persons and the protected data, the name of the data type shall be used as a replacement text [8], e.g., if there are "i" number of plaintiffs mentioned in the decision it should be replaced as $Plaintiff_1, Plaintiff_2,...Plaintiff_i$.

The law also states that all data enabling the identification of a natural person, a legal person or an organization without legal personality have to be removed. Nevertheless, the exact range of data to be anonymized is not defined.

Recital 27 of GDPR (gdpr-info.eu/recitals/no-27/) (accessed on 1 May 2021) states that the Regulation does not apply to the personal data of deceased people and leaves this question to the EU Member States. In Hungary, the Act CXII of 2011 (InfoAct) states that the data subject's rights after their death could be exercised either by a person appointed by the data subject during their life or a close relative (www.twobirds.com/en/in-focus/general-data-protection-regulation/gdpr-tracker/deceased-persons) (accessed on 1 May 2021).

*5.2. Criticism of Current Regulation*

Although the current regulation tries to embrace the rules defined by the GDPR, some points can be a subject of debate.

First of all, it is important to point out that the Hungarian law does not distinguish direct and quasi-identifiers. This is problematic, since one could easily associate only with direct identifiers and we will see that this is the case when it comes to practice. With quasi-identifiers remaining in texts, the parties could become easily identifiable [40].

It is important to remark that mentioning a rare event has to be considered as a quasi-identifier (sometimes as a direct identifier), but removing or modifying this can easily contradict the commandment stating that the sensitive data have to be modified in a way that they do not change the established facts of the case [7,8]. Nevertheless, in these cases, generalization of the events could be a possible solution for the contradiction.

Another remark about the law would be that it requires the decisions to be published within 30 days. Although the year when a case started can currently be determined from the case identifier and the end of the case is also mentioned in the document, the results on the Netflix dataset may serve as a basis for revising this condition [20,21].

*5.3. Current Practice and Potential Risks*

The practice of anonymization on court decisions shows that the names of the parties and all their data (address, mother's name, date of birth etc.), so the direct identifiers have been completely removed or in certain cases people's full names or company names have been replaced with their monograms.

Consequently, the texts usually contain a remarkable amount of quasi-identifiers. Generally, every attribute that can be used to perform a linking attack, is a quasi-identifier. Some examples would be: sex, age, locations, professions, monograms, dates (day, month and

year), company types, activities of a company. The name of judges or attorneys involved could provide further links, e.g., the name of the lawyer, even if all data related to location has been removed/replaced in the document, could provide additional information on the location of the events, especially if the lawyer is placed in the rural area.

### 5.4. Case Studies

In order to present the power of quasi-identifiers and also to identify some of the potential quasi-identifiers in the legal domain, research has been conducted using publicly available legal cases in Hungarian.

### 5.4.1. Datasets and Search Framework

Our case study was conducted by using the LEXPERT service (https://www.lexpert.hu) (accessed on 1 May 2021), which is a self-developed online database of the freely available Hungarian legal cases. LEXPERT ensures an Elasticsearch-based search platform for the legal documents and links the connecting documents and law references. The following documents are included in this database:

- Case-law of the Constitutional Court: decisions and procedural decisions of the Constitutional Court of Hungary, currently approximately 9800 documents.
- Court Decisions: consists of decisions from all tiers of the Hungarian judicial system from district courts up to the Supreme Court. This is the biggest dataset containing approximately 160,000 documents. This study was conducted using this dataset.
- Uniformity Decisions: binding decisions of the Supreme Court in uniformity cases to ensure the uniform application of law within the Hungarian judiciary, currently containing approximately 200 documents.
- Selected case-law: selection of the most important court decisions, currently containing approximately 2200.
- Division opinions and decisions: decisions of the divisions of the courts on abstract, theoretical legal questions. Contains approximately 400 documents.

### 5.4.2. Illustrative Examples

One of the simplest cases was when a judge started a labour lawsuit against their employee. Since names of attorneys and judges do not have to be removed, in this case, their workplace and their name remained in the text.

We have noticed that many documents contain exact dates, which is a possible source of a serious information leak. Sometimes, not only the year, month and day are mentioned but the hour and minute as well. When these dates refer to the general flow of a legal case (e.g., dates of previous decisions, appeals etc.) it usually does not mean a potential risk.

However, when a date refers to a unique event (e.g., someone was gored by a bull, died during a routine surgery, is starting a business, etc.), it can pose a serious threat for re-identification, since these data usually appear in local newspapers or can be looked up in the Hungarian Company database or other publicly available databases. The death date is handled differently in the different EU member states, for example, in Hungary it is not personal data; however, in Denmark it remains personal data after the death of the person [78].

An example for the possible threat that multiple quasi-identifiers may pose would be the following case. In southern Hungary, there was a case when a then-92-year-old deceased person's date of death and age were published. The authors of [79] showed that this is sensitive information, because this age can be applied to no more than the 0.5 percent of the population. It is known that, at that time, around 4000 women lived in Békés county with over 85 years of age, from the total population of 397,791 [80]. This information cannot be sensitive because this represents one percent of the total population, and if we did not know the sex of the victim, the age could refer to nearly the 1.5% of the population. After a subtraction, we know exactly her age when she died and there were no more than 300 people. Of these, 200 were woman in that county and were 92 years old; this is the 0.05% of

the total population, which is significantly lower than the recommended minimum (0.5%). With only two pseudo-identifiers, we were able to narrow down the number of potential individuals to around 200. In this case, we did not even use the other pseudo-identifiers that could be learnt from the text, the exact date of the death and where she lived. From the text, we know that the woman lived in a small town with no more than 5000 inhabitants. This means if the old people are distributed uniformly in the examined area, that there were four different people at that time fulfilling these criteria. The shrinkage of equivalence class sizes can be followed in Figure 6. This information can be paired with the local journal's obituary section, easily identifying the dead person.

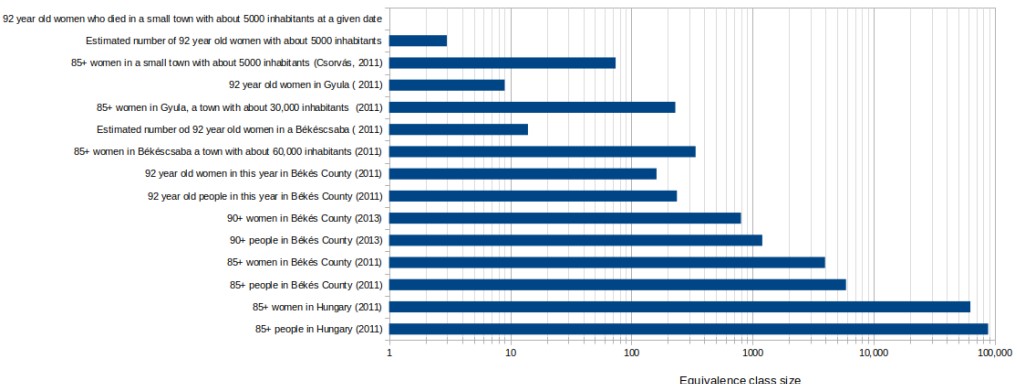

**Figure 6.** The image shows the risk if the age and the gender of an old person (85+) is published in a legal document. The different bar plots shows the equivalence class sizes, when the document does not contain or contains information about settlement involved [80].

Naturally, the de-anonymized data could be used for linking entities further.

Another potential risk factor is that the documents related to the medical field often publish the full medical history of a person with exact dates and types of surgeries, drugs taken, etc. By themselves, these data could be used to reveal someone's identity; however, linking these to public databases can be difficult. The problem is that this information can be sensitive. We have found a case where only the case of medical malpractice was enough to identify both parties with a simple Google query. It made the matching process easier that the dates were also mentioned in the document. In this example, the whole medical history of the patient was mentioned referring to sensitive types of surgeries and medical treatments.

Another date that appears in each document is connected to geolocation, namely, the courts involved. Since the population density is not equally distributed across the country (which is likely the case for the majority of the countries in the world), a court operating in a smaller county can be considered as a quasi-identifier. For example, if there is a hospital involved in a case and the type of surgery is mentioned, in many cases, the name of the institute can be easily re-identified. If any name of a settlement accidentally remains in the text, the re-identification can be even easier. For example, a particular issue in the Hungarian language is that when a name of a settlement is referred as "something from Xy settlement", the name of that settlement is not written by capital letters but lower-case letters. The human annotators tended to miss these data.

Another quasi-identifier would be when a natural or artificial formation type (e.g., reservoir, lake, cave, river, mine etc.) is mentioned in the text even if the name of it is completely removed. Since the region of the case is known from the court, these formations can be unique identifiers. There was a case where the starting characters of the settlements involved remained in the text, alongside the term reservoir of the river XY. Since there is only one of this, the settlements could easily be identified. The text also contained fragments from parcel numbers, thus the owners could be identified from these data.

The profession could be another quasi-identifier that can seriously reduce the equivalence class size, hence increasing the risk of de-anonymization. For example, there was

a case in which academic members were involved. In itself, this information reduces the equivalence class size to around 300. The case mentioned the scientific domain alongside the monograms of these people, giving more than enough information to de-anonymize the parties.

In case of companies, the scope of activities could also be a potential identifier. Even if this information would not identify a company uniquely, knowing the date when it started and in which city would provide enough information to do so, as we found in another criminal case. Since the members of one company can be listed even historically in the Hungarian Company Database, it is relatively easy to find the accused people. In this case, many fragments of bank account numbers were also available alongside the name of the banks, providing additional information.

Using a good name entity recognition together with a non-appropriate anonymization technique can be a double-shaped blade. There was an example case in the published data, where a masking technique anonymized the hospital's name as "U... Rt.". From this masked name, we have three pieces of information. First, the presence of the "..." refers to the fact that these data are a direct identifier. Secondly, the medical treatment was made in a private hospital because "Rt" means public limited company, and we know the company's starting letter. There were only two different medical service companies in Hungary at that time, which name starts with U, but only one of them is authorized to perform such medical treatment. However, if the name of the hospital had only been generalized or suppressed perfectly, we could have not looked this information up in the Hungarian Company database.

We have provided some examples of how only the company type, the location of the company, and the date of registration can reduce the equivalence class sizes. The results can be seen in Figure 7. Knowing these three pieces of information reduces the equivalence class sizes so drastically that only Budapest could make this value above 10,000 equivalence class size compared to the whole country. Even in the case of the cities with the largest population after Budapest (Debrecen, Miskolc, Szeged, Győr), the size of the equivalence class reached, at most, 400.

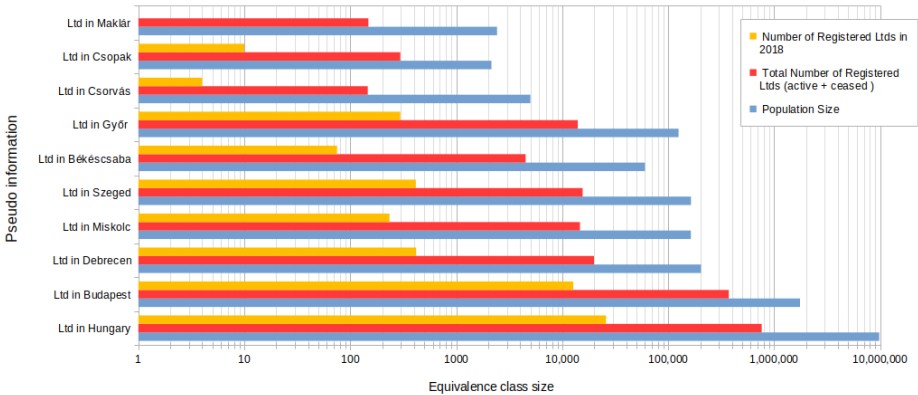

**Figure 7.** The image shows the risk if the company form is published together with the date of the registration, in the case if the headquarters is known or unknown from the corpus. The blue bar shows the total number of the registered active or inactive Ltds in Hungary or in the given city. The orange bar plots those Ltds, which are registered in 2018 [80].

To sum up the results, it can be stated that the type of quasi-identifiers is broad not only in domain (from dates to natural formation types) but also in their nature (simple nouns to chain of events).

The risk of re-identification dramatically increases when the case can be connected to time and location, and all documents are connected to a court, and it has been year at least since the case started, as these data must be published according to the current regulation. Generally, following common sense, all data, which is rare in some sense, increases this

risk. The cases highlighted in this section showed that mentioning exact dates and starting letters should be avoided, since these additional pieces of information reduce equivalence class sizes drastically. Nevertheless, the risk of re-identification should be estimated by involving as many quasi-identifiers as possible from the given text since considering these data together, there may already be enough information for de-anonymization. The question arises: how could be this risk quantified? We are pursuing the answer for this question in the following section.

*5.5. Quantifying Risk*

Intuitively, it can be stated that not all quasi-identifiers have the same "strength", "danger-factor" or the same information content as far as de-anonymization is concerned.

In Information Theory, the metric for measuring information is entropy. Shannon Entropy has been defined in Definition 9.

**Theorem 9** (Shannon Entropy [81])**.**

$$H_S = - \sum_{n=1}^{N} p_i \cdot \log_2 p_i \qquad (2)$$

*where $P = (p_1, p_2, ...p_N)$ is a discrete probability distribution and $p_i \geq 0$ and $\sum_{n=1}^{N} p_i = 1$*

Entropy is measured in bits. Assuming that we have an equivalence class with size N and all members of this class occur with the same probability, by picking randomly, on average, $log_2 N$ guesses have to be made to find a certain element from that equivalence class and $log_2 N$ is equal to the Shannon Entropy. If the entities from an equivalence class do not have the same probability for being the element to be found, the entropy will be lower; in case when it is known, $H = 0$ stands. This means that a theoretical maximum entropy can be calculated, assuming that the attacker is performing random picking on the equivalence class.

The size of an equivalence class could be gathered from demographic statistics or other publicly available data sources such as company databases, medical databases, voters registration list and so forth.

In case the exact size of the equivalence class characterized by the extracted set of quasi-identifiers is not known, by calculating conditional probabilities or applying Bayes' Theorem [82] it is possible to estimate the size of this equivalence class.

**Theorem 10** (Conditional probability [83])**.**

$$P(A/B) = \frac{P(A \cap B)}{P(B)} \qquad (3)$$

**Theorem 11** (Bayes' Theorem [82,83])**.**

$$P(A/B) = \frac{P(A) \cdot P(B/A)}{P(B)} \qquad (4)$$

However, the easiest solution would be to assume that $A_1, A_2, ...A_i$ probability variables are independent to each other, and the conditional probabilities of these attributes can be calculated as: $P(A_1 \cap A_2 \cap ... \cap A_i) = P(A_1)P(A_2)...P(A_i)$. Although the independence is not true in many cases, this simple equation could be easily used as an estimation in many cases.

Hence, by extracting and linking attributes to a specific person and estimating the equivalence class size of all these attributes occurring together, the entropy can be estimated as well. However, it is important to point out that by performing the estimation using more parameters, the level of potential error also increases, and with it the variance of potential

information content also increases. For instance, in two cases having equal equivalence class sizes but one has been estimated using two attributes and the other has been estimated using four, it is expected that the latter scenario can be riskier in terms of de-anonymization.

Figure 8 shows the connection between entropy and risk.

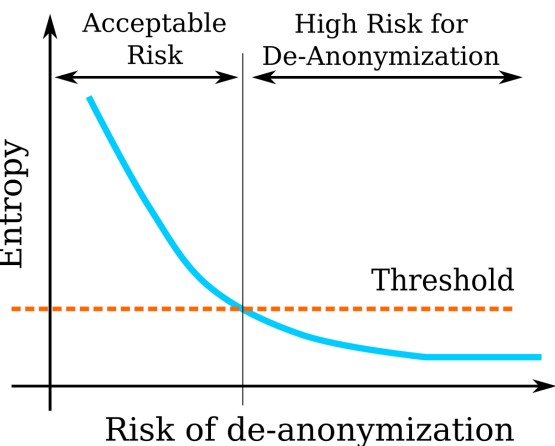

**Figure 8.** Connection between entropy and risk.

According to our understanding, the risk as function of entropy is such that by decreasing entropy (by knowing more of a specific person) the risk increases.

Since there is no case when the probability of de-anonymization is zero, the de-identification process must aim to raise the entropy of the set of quasi-identifiers above a certain threshold.

*5.6. The Threshold*

As Figure 8 suggests, there is a threshold dividing the acceptable and high-risk regions. The question is, then, what is a reasonable choice as a threshold value? The answer is, it depends on the actual application. The authors of [34] suggested a weighting for attributes based on the count of non-zero elements of an attribute and define a condition for attribute sparsity (Definition 13). Since, in our case, the real size of the equivalence class usually cannot be determined using the set of legal documents but requires other publicly available databases, the definitions have been modified accordingly.

**Theorem 12** (Weight of an attribute [34]). *"A weight of an attribute is $w_i = \frac{1}{log_2 N}$ where N is the size of the equivalence class".*

**Theorem 13** (t-sparsity [34]). *"An attribute is t-rare if $w_i = \frac{1}{log_2 N} \geq t$ where N is the size of the equivalence class and t is a threshold value, $0 < t \leq 1$".*

In the Netflix case study in [34], $t = 0.07$ and $t = 0.075$ have been used as sparsity values, suggesting approx. 20,000 and 10,000 equivalence class sizes, 13.33 and 14.29 bits of entropy, respectively.

In [79], a rule of thumb has been mentioned in terms or equivalence class sizes: 0.5% of the population that would mean 48,850 in Hungary (approx. 15.6 bit or entropy). The presentation has mentioned examples, such as settlements under 10,000 population or the population above the age of 85 being considered dangerous, hence an anonymization method (generalization or suppression) has to be applied in these cases.

To provide some examples, entropy values and weights for attributes have been calculated and presented in Table 1. Surprisingly, 33 bits of information are enough to identify someone uniquely from the world's population [33].

**Table 1.** "33 bits of entropy are sufficient to identify an individual uniquely among the world's population" [33].

|  | **H (bit)** | **w (1/bit)** |
| --- | --- | --- |
| World's population | 32.86 | 0.03 |
| Hungary's population | 23.22 | 0.04 |
| Budapest's population | 20.74 | 0.05 |
| Age 85+ | 17.61 | 0.06 |
| Medical doctors | 15.33 | 0.07 |
| Professional football player | 10.19 | 0.10 |
| Nr. of companies | 20.28 | 0.05 |
| Nr. of Ltds | 19.54 | 0.05 |
| Nr. of Ltds founded in 2018 | 13.63 | 0.07 |
| Nr. of Ltds founded in Jan 2018 | 10.26 | 0.10 |
| Hungarian Academy of Sciences (HAS) member | 8.15 | 0.12 |
| Member of HAS in Engineering | 4.91 | 0.20 |

From the point of view of the data owner, it is crucial to know or at least to estimate the information gain when an additional attribute is known. This information gain appears on the attacker's side.

**Theorem 14** (Information gain [83])**.**

$$IG(Y/X) = H(Y) - H(Y/X) \tag{5}$$

*where IG is the information gain on the event Y if X is given, H(Y) is the entropy of event Y and H(Y/X) is the conditional entropy of the event Y given event X.*

**Theorem 15** (Conditional entropy [83])**.**

$$H(Y/X) = \sum_{x \in X, y \in Y} p(x,y) \cdot log(\frac{p(x,y)}{p(x)}) \tag{6}$$

*alternatively,*

$$H(Y/X) = \sum_{j} P(X = v) \cdot H(Y|X = v) \tag{7}$$

*where H(Y/X) is the conditional entropy of the event Y given event X, P(X = v) is the probability of event X taking the value v, H(Y | X = v) is the entropy of event Y if event X takes the value v.*

Denote by X and Y that there is an additional attribute and there is a person in the legal text. In legal texts, usually, not only the attribute X is mentioned, but so is a certain value of it, denoted by v. If it is possible, $H(Y/X = v)$ has to be calculated to see how the additional information decreases the complexity of the given problem. In case it is not possible, the entropy of event X = v can be estimated by considering event X generally, since $H(Y/X = v) \leq H(Y/X)$. In a worst-case scenario for the attacker, if we consider the probability distribution of X as uniform, the conditional entropy $H(Y/X)$ will have the highest value of all other distributions. It can be seen that from the data owner's point of view this is the best-case scenario so this is a weak estimation. Despite that, this estimation still can be useful, since, if in the worst-case scenario for the attacker, the entropy of the problem decreases under the threshold value, it can be stated that an attacker is likely to have enough information to re-identify data. Another possibility for estimation would be, when the probability distribution of event X is not uniform, but the most probable values are known with their probabilities. In this case, the minimum entropy would be $H_{min}(X) = -p_i \cdot log_2 p_i$ where i denotes the most probable element of the distribution. It is clear that in this case $H_{min}(X) \leq H(X)$.

In Table 2, some examples have been provided to show how much a given problem could be simplified by a piece of additional information. The calculations assume that both the problem and the additional information (e.g., the monograms) are uniformly distributed. Despite the fact that the Hungarian alphabet is 44 letters, we took that into account with only 40 letters during the calculations, not counting the letters q, w, x, y, which are extremely rare in the Hungarian language as starting letters. During the calculations, the full English alphabet was taken into account.

If the entropy of a problem is known and it can be estimated that how much a piece of additional information reduces the fraction of the equivalence class size, the entropies can be just subtracted from each other since $log_a(b/c) = log_a b - log_a c$.

If there is a case mentioning the two-letter monogram of a witness and provides the additional information that he/she is an academic member, it is highly probable that the person can be identified, since knowing that someone is an academic member means 8.15 bits of entropy and subtracting 10.64 bits for knowing the monogram would result in negative entropy, and zero entropy would mean that the person has been identified. When we consider the difference between the entropy of Nr. of Ltds founded in 2018 and Nr. of Ltds founded in January 2018 from Table 1, we can learn that the difference is 3.37 bits, which is close to the value of 3.58 bits potential information gain presented in Table 2.

**Table 2.** Possible information gains when additional information is known.

|  | Information Gain |
| --- | --- |
| Sex | 1 |
| Month, when year given | 3.58 |
| 1 year from 30 years range | 4.91 |
| Year and month in 30 years range | 8.49 |
| Year, month day from 30 years range | 13.42 |
| Monogram Hungarian (1 letter) | 5.32 |
| Monogram Hungarian (2 letters) | 10.64 |
| Monogram Hungarian (3 letters) | 15.97 |
| Monogram English (1 letter) | 4.70 |
| Monogram English (2 letters) | 9.40 |
| Monogram English (3 letters) | 14.10 |

## 6. Automatized Workflows for Pseudonymization

Using an automatized named entity recognition workflow is not enough to comply with the GDPR. The importance of different pseudo identifiers must be considered during the anonymization workflow. The available automatized pseudonymization applications should be improved using different methodologies such as event recognition, semantic role labeling and risk analysis to create effective pseudonymization tools (Figure 9). In this section, the parts of the process are described.

Given a legal text, finding direct and as many quasi-identifiers as possible is the first step towards the pseudonymized legal document. This is carried out by recognizing these entities from the text. This process is also known as Named Entity Recognition (NER) [84–86]. There is a vast number of anonymizer solutions that are based on NER models [2,25,58,66,67,87–89]. As a consequence, the performance of the NER model used in any anonymization architecture highly influences the performance of the anonymization solution [25]. However, it is important to note here that finding direct identifiers is a necessary but not a sufficient condition for anonymization [14–16]. Finding pseudo-identifiers is also important, but the range of quasi-identifiers can be wide due to the nature of legal texts. This is because a part of a legal case, namely the matter of fact, describes the whole story of the parties in detail, and, usually, this part is full of quasi-identifiers that are hard to discover automatically. In practice, finding all types of quasi-identifiers is a very difficult task and the selection of quasi-identifiers should be preceded by a careful risk analysis.

**Figure 9.** Named Entity Recognition should be extended with Named Entity Linking, Event Recognition and Semantic Role Labeling to realize a GDPR compatible pseudonymization framework for legal cases.

In contrast to medical records, at this stage, the extracted data cannot be put into tabular format to perform risk analysis. This is because medical health records generally contain information about one person but this is not the case in a legal document. Usually, there are two sides (plaintiff and defendant) and there could be more people involved on each side. The case could also mention other people involved indirectly, e.g., witnesses, doctors, visited places.

To solve this issue, the extracted name-typed named entities (e.g., name of person or institute) have to be connected to the other extracted entities (dates, age, profession, etc.). One possible solution to perform such identification of connections would be via dependency parsing. Once entities have been extracted, wherever possible, an equivalence class size has to be assigned to each entity, either by using a knowledge base or by looking up tables of statistics. This part of the process is denominated as Named Entity Linking [90] in Figure 9.

Since the importance of events as quasi-identifiers has been emphasized in this study, the events have to be extracted from the texts alongside the arguments of these events. This process is known as Event Extraction [76,77,91–93].

The task of specifying "who did what to whom where and why" from text is called Semantic Role Labeling [94]. It is important to point out that from the anonymization point of view, classification of events based on their rarity is more important than finding all the answers to the questions mentioned above. Nevertheless, these answers could lead to a better rarity estimation.

As a result of this stage, all extracted information can be stored in a matrix where each row refers to a name-typed entity and each column to a specific attribute of this entity. Motwani and Nabar have shown that it is possible to transform unstructured data into a relational database format containing only zeroes and ones, which is sparse [24].

Once these data have been collected, the risk analysis is the next task. By risk analysis, we mean estimation of equivalence class sizes of each attribute connected to a specific named entity (e.g., Person type) by using knowledge bases and/or demographic statistics, or third-party databases. In the case of events, the focus is on estimating rarity similarly to the non-event type entities. From these data, the risk could be estimated by calculating entropy values for each data extracted from the text and, more importantly, to the collection of these data and comparing them to a given threshold.

The next step is transforming the data. This step is the application of anonymization techniques such as generalization, masking, slicing, suppressing, etc., on the extracted data. By these techniques, the risk of re-identification is decreased. The modified entities then have to be put into the document, replacing the extracted forms.

As a final step, the validation of the anonymized document is made from two aspects. On one hand, the likeliness of re-identification and, on the other hand, the intelligibility of the document. This is important in legal decisions, since, for example, if every entity were replaced by "..." characters, the text would become confusing due to the relatively high number of participants in a legal case.

Should the anonymized text fail on validation, the whole procedure is repeated from the risk analysis step until the termination condition is met.

*About the Feasibility of a GDPR Compatible Automatized Pseudonymization Framework*

By using the technologies suggested in Section 6, the chances of identifying the people involved in a case can be significantly reduced. To achieve a GDPR-compatible anonymization process, which means that all of the involved people have been de-identified, each step (Named Entity Recognition, Event Recognition, Semantic Role Labeling, etc.) must work with very high efficiency. We can think it over in the following example, which considers only the importance of the NER from the tools as mentioned earlier. If one has a NER model that recognizes the sensitive entities with 99% accuracy, and there are 20 entities in the text, which should be identified, the probability that at least one entity (quasi-identifier) will remain in the document is still significant: $1 - 0.99^{20} = 18.2\%$. This is why one of the pillars of the TILD methodology [27] is to test the anonymization systems via motivated intruder testing involving humans [95]. Most of the state-of-the-art pseudonymization tools use the NER to create a pseudonymized document only. This does not ensure that some person can be re-identified after some specific event. On the other hand, if all of these entities are replaced, these entities should be replaced with some other words that fulfill the grammatical role of the original text in order to preserve the information in the content and the clarity of the text. This justifies that a good semantic analysis can improve the quality of the pseudonymization process.

The conclusion is that data owners have to accept that legal cases may not be fully anonymized, only pseudonymized with an acceptable risk [33,34,49]. Data owners can reduce the chance of a successful attack by conducting a risk analysis and applying the proposed pseudonymization technologies presented in this section. The existing risk analysis methodologies are based on databases, where the distribution of the information is symmetrical, such as medical databases, where every record has the same properties. Legal documents are unstructured sources of possible quasi-identifiers. The database built from the linking documents is a large, asymmetrical dataset, which should be considered to create different and more effective risk analysis and pseudonymization algorithms.

## 7. Conclusions

The digitalized, openly accessible court decisions have a fundamental role in improving the decision-making processes and making the administration of judicial systems more transparent. The GDPR forms strict regulations for openly publishing private data. Therefore, the published data of the mentioned parties of the court decisions should be pseudonymized. During the pseudonymization process, the direct and indirect identifiers are masked, generalized, or replaced by taxonomies. The main difference between anonymization and pseudonymization is that the latter process is reversible. However, anonymizing the data can often destroy the utility of the published data.

Many automatized solutions have been developed in the different EU member states to accelerate the solution of this process. Most of these tools use modern named entity recognition-based methods to classify, mask or generalize the direct identifiers. Therefore, the solutions mentioned above pseudonymize these documents only. Nevertheless, these tools do not take risks and the utilizability of the pseudonymized data into consideration. The legal documents are unstructured texts where the information-loss after removing the different parts of the sentence should be considered. Privacy-preserving publishing can be achieved by the application of differential privacy algorithms, as in the case of public health data. However, the structure and the information content of a legal case greatly differs from health records, where the same type of data represents every individual. In legal documents, a wide range of attributes of different kind are available, referring to the involved parties. Moreover, legal documents can contain additional information about the relations of the involved parties and rare events. Hence, the personal data can be represented by a sparse matrix of the attributes. It has been shown that this kind of anonymized data is inherently prone to de-anonymization.

Therefore, a named entity recognizer tool is essential to make a fair anonymization process, but it is not enough. Named entity recognition, event recognition, semantic role

linking, named entity linking should be used together with the anonymization algorithms (k-anonymity, l-diversity, etc.) to quantify the level and the utility of the process. The risk analysis can be performed by using statistical methods and entropy by estimating equivalence class sizes. To sum it up, due to the No Free Lunch Theorem [68–70], there is no single easy solution that exists for anonymization that works for all approaches in all possible scenarios.

**Author Contributions:** Conceptualization, T.O.; methodology, G.M.C. and T.O.; software, G.M.C., D.N., J.P.V. and R.V.; resources, D.N. and J.P.V.; writing—original draft preparation, G.M.C., R.V., D.N. and T.O.; writing—review and editing, D.N. and T.O.; visualization, G.M.C. and T.O.; supervision, T.O.; project administration, D.N.; funding acquisition, J.P.V. All authors have read and agreed to the published version of the manuscript.

**Funding:** Project No. 2020-1.1.2-PIACI-KFI-2020-00049 has been implemented with the support provided from the National Research, Development and Innovation Fund of Hungary, financed under the 2020-1.1.2-PIACI KFI funding scheme.

**Institutional Review Board Statement:** Not applicable.

**Informed Consent Statement:** Not applicable.

**Conflicts of Interest:** The authors declare no conflict of interest.

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
