# Peer review of "Challenges and Open Problems of Legal Document Anonymization"

_symmetry, doi:10.3390/sym13081490_

Round 1

Reviewer 1 Report

This paper central aim is to review and highlight the open problems and the possible methods for anonymizing legal documents. The authors considers the general use case of the Hungarian justice judiciary system, which accepts several types of documents that contains personal data. This is even complicated further by the mandatory consideration of the European GDPR regulation. At first, the reader is let to believe that the work aims to be a survey. Nevertheless, the paper does not respect the general structure of a survey paper, as it does not put much effort into discussing in a constructively critical manner the presented contributions, and the authors attempt to propose a personal view on the matter, which is fine by itself. Nevertheless, the paper is placed in the middle between a survey paper, and a genuine contribution-reporting article. This is the main issue of the manuscript. I would say that the following aspects should be corrected before the paper may be taken into consideration.

  1. The authors should clearly decide whether they propose a survey work, or a contribution paper. If it is the latter, then the use case should be fully described, the data set must be presented, and the assessment and testing infrastructure must be described, including the architecture of the software system, and some considerations concerning the hardware. This is the general way of reporting such a contribution.
  2. If the paper is a survey, then it should blend more with the respective style.
  3. The English language should be proofread and improved.

Author Response

Please, check the attached file.

Reviewer 2 Report

The paper could be accepted in the current form

Author Response

Dear Reviewer,
Thank you very much for the positive feedback.

Reviewer 3 Report

I have completed the paper review and identify the following shortcoming in the paper.  

1- The paper surveyed the existing work and provide a comparative literature review about existing practices in research. 

2- the paper does not add significant knowledge to the research  

3- I do not see any criticism on exiting work to provide new guidelines or policies for legal document privacy protection.  

4- There are many spelling mistakes "New Zeeland" and undefined abbreviations ("GDPR" in the abstract) used in the paper which should not be the case.  

5- The authors need to set their priorities straight. either author is drafting this paper as a survey paper, or the author are proposing new polices and mechanism to deal with a legal document. 

Author Response

Please, check the attached file.

Reviewer 4 Report

The article ‘Challenges and Open Problems of Legal Document Anonymization’ as the title suggests, presents the challenges and problems related to data anonymization in legal domains. Considering the decisions from publicly available of Hungarian Register of Court decisions, the authors examine these data anonymization challenges. They also propose a small pipeline of pseudonymization process for legal documents.

The article is well written and with clear comparisons between medical and legal domains (section 5), the readers who are not familiar with legal domains can understand the challenges and problems related to anonymization in legal domains.

However, I feel that the article can be further improved. The authors have quickly proposed a pseudonymization process in section 7. The goal of this section is not clear to me. Is it a novel contribution or an improvement of the existing processes? Considering the title of the article, I wonder the purpose of this section.

Though the authors have clearly defined the various important terms in their article, I feel that the terms anonymization, pseudonymization needs to be redefined with respect to the legal domain.

Line 213: ‘This is the same as l-sensitive k-anonymity.’  is not clear to me. Do the authors mean p-sensitive k-anonymity (discussed just above, where p=l).

Line 245-246: if the distance between the distribution. What does distance mean here?

I also feel that there are a lot of repetitions in section 6. To give an example, the instances related to (extraordinary) events are repeated multiple times.

The authors may need to clearly explain the purpose and scope of this article in the introductory section, especially considering the title. For example, my above remark on section 7.

Minor remarks

  1. Line 26-27 is repetitive. Please check line 24-25. On the other hand, these openly accessible legal documents can contain much sensitive information.
  2. Table 1: New Zeeland -> New Zealand
  3. Line 382-383 modifying them soonly performing Named Entity Recognition and modifying the extracted entities. Not clear
  4. Line 396: good tactics -> good tactic
  5. Line 418 : , the decision have -> , the decisions have
  6. Line 419: In both cases the -> In both the cases, the

Author Response

Please, check the attached file.

Round 2

Reviewer 1 Report

Considering the first version of the authors' paper, which I reviewed, I appreciate that the authors demonstrated their willingness to improve the paper, and implemented most of the required changes. The paper may be regarded as a survey work, although not an extended one, which is illustrared with examples from the Hungarian legal system. As additional suggestions, the authors should highlight and justify better the paper's link to the scope of the journal, and they should proofread the paper's text one more time.

Author Response

Thank you very much for your valuable feedback. The paper has been proofread and the required changes were denoted by cyan colour.

The topic of this paper is related to the two keywords of the following special issue: "Symmetric and Asymmetric Data in Solution Models, Part II":

  • Symmetric and asymmetric risks
  • Description of symmetric and asymmetric data in models. 

The existing risk analysis methodologies are based on databases, where the distribution of the information is symmetrical, like medical databases, where every record has the same properties (e.g., name, age, illness, treatment, etc.). Legal documents are unstructured sources of possible quasi-identifiers. The database built from the linking documents is a big asymmetrical dataset, which should be considered to create different, more effective risk analysis and pseudonymization algorithms.

The above answer was inserted into section 6.1. 

Another explanation inserted into section 4:

The right side of Figure 5. illustrates the case of a typical medical database, where every row contains the same information about an individual. This database is symmetrical because every record has the same identifiers, and we know every possible quasi-identifier in this task. The health data anonymization methodologies use this symmetry or structural regularity. In contrast, the legal documents can be very asymmetrical, hard to find similar structural regularities, which increase the complexity of the risk analysis of these models. 

The most obvious example is the role of the personal data of deceased persons in the documents. GDPR leaves the handling of such data to the EU member states. Hence some countries do not care about the personal rights of dead persons. However, in probate proceedings, some sensitive data of the dead person can be published. If we know the link between the deceased person and the plaintiff, who is, e.g. the only living relative, we can re-identify his data. A more general risk for the asymmetrical dataset is that the document contains three-three quasi-identifiers for two individuals, which are insufficient to re-identify the involved persons. However, these six identifiers can belong to six different categories (like occupation, age, etc.), and by knowing the relation between the individuals can be possible to re-identify them.

Reviewer 3 Report

The authors are required to extend section 3. Highlighting two or more privacy concerns that are specifically identified in the judicial system from the literature.

Section 2 and 4 need to be reduced and marge together. The authors used explained all the existing approaches that are not needed/ used in this paper. The authors should focus more on algorithms that directly or indirectly impact the legal document before and after the anonymization process.

In section 7, the authors are required to provide subsections that demonstrate the feasibility of GDPR compatible pseudonymization framework with the legal system and provide the conclusive remarks/ guidelines for readers as a takeaway.

Authors are advised to get a native English speaker for paper review/proofreading.

Reviewer 4 Report

I would like to thank the authors for taking into consideration my previous review comments and submitting a significantly updated article. They have updated the abstract clarifying the contributions of their work. The introduction has also been updated, especially with necessary references for the readers who wish to further understand various aspects of anonymization. Section 6.4 is updated with a dedicated subsection on the data and the application used for the study.

Author Response

Thank you for your positive feedback.

Round 3

Reviewer 3 Report

Good job authors.